# Impact of Skin on Microwave Tomography in the Lossy Coupling Medium

**DOI:** 10.3390/s22197353

**Published:** 2022-09-28

**Authors:** Paul Meaney, Shireen Geimer, Amir Golnabi, Keith Paulsen

**Affiliations:** 1Thayer School of Engineering, Dartmouth College, Hanover, NH 03755, USA; 2AI4ALL, San Francisco, CA 94104, USA

**Keywords:** microwave imaging, breast cancer, finite element, finite difference time domain, skin effect, skin thickness

## Abstract

In microwave imaging, the effects of skin on recovering property distributions of tissue underneath the surface may be significant because it has high dielectric contrast with subcutaneous fat, which inevitably causes significant signal reflections. While the thickness of skin, especially relative to the wavelengths in use, would presumably have minor effects, it can introduce practical difficulties, for instance, in reflection-based imaging techniques, where the impact of the skin is large—often as high as two orders of magnitude greater than that of signals from underlying tumors in the breast imaging setting. However, in tomography cases utilizing transmission-based measurement data and lossy coupling materials, the situation is considerably different. Accurately implementing a skin layer for numerical modeling purposes is challenging because of the need to discretize the size and shape of the skin without increasing computational overhead substantially. In this paper, we assess the effects of the skin on field solutions in a realistic 3D model of a human breast. We demonstrate that the small changes in transmission field values introduced by including the skin cause minor differences in reconstructed images.

## 1. Introduction

Accounting for the effects of skin on electromagnetic (EM) wave propagation imposes important challenges for both microwave imaging and therapy [1]. Its associated properties are high relative to adjoining tissue—primarily subcutaneous fat—largely because of vascularity and water content compared with fattier tissue [2]. In addition, incident fields generally impinge on skin from outside the body where the medium ranges from air to coupling materials such as saline, glycerin, oil, etc. [3,4,5,6]. The net effect often results in EM reflections and excitations of associated standing waves which can undermine effectiveness of EM imaging or therapy. Even though the skin is generally thin—typically 1–2 mm—its electromagnetic impact can be significant.

The impact of the skin is particularly notable in reflection-based, time-domain microwave radar [1]. Here, the coupling medium is either low loss or even air, and as a result, reflections from the air-skin and skin-fat interfaces are large, often orders of magnitude greater than reflections from internal scatterers such as a tumor [6]. When the imaging algorithm is fully or partially model-based, skin effects can be subtracted as long as its geometry is known. Simulation studies have demonstrated good recovery of internal tumor when skin properties and geometry are known [7], for example, from optical scanners integrated directly with microwave imaging instruments [8,9,10]. However, these scanners provide coordinates of the skin surface—not its thickness [8,10], and the extent to which surface information is sufficient for modeling the skin when its thickness is only marginally greater than the measurement errors is unknown.

Transmission-based techniques are the focus of this paper [11,12]. These methods have typically ignored the presence of skin [13,14,15] under the assumption that skin effects area minimal, in part because of use of relatively high permittivity and very lossy coupling media—to minimize surface waves and associated multipath signals [16]. Testing for the effects of skin in simulations and experiments is difficult, and not possible in patients. Skin phantom materials are only recently becoming available but are challenging to apply uniformly around a phantom [17]. Phantoms constructed by 3D printing the breast shape with separate compartments for tissue-mimicking dielectric liquids have been created from MR scans of actual breasts, but they generally do not include the skin layer [18]. In fact, the plastic layers used to separate different tissue types (adipose, fibroglandular and tumor) are sufficiently thick that they generate their own scattering patterns that detract from the intended ground truth [19]. Generating realistic skin models is nontrivial even in simulations because of the discretization required to produce accurate solutions over the entire breast volume [20]. While finite element methods can utilize nonuniform mesh spacing to represent curved breast geometries more efficiently than lattice-based FDTD techniques, the overall problem size is still daunting. Nonetheless, implementations of skin models have been reported in which the details are imported directly as part of the breast geometries and properties from the University of Wisconsin database [21,22,23,24,25,26,27,28,29].

This paper presents a framework for assessing the impact of skin on microwave transmission signals. It describes differences between dielectric property distributions represented in terms of elements, or conversely, in their nodes, and the influence of skin. For simplicity, 1D representations are used to illustrate the impact of skin more clearly. Results show field distributions for 3D cases mapped to 2D microwave property distributions with and without skin and confirm previous assumptions that the skin layer has modest impact on microwave imaging results when lossy bath/transmission-based methods are used, especially when compared to the impact of skin encountered in radar-based, time domain approaches.

## 2. Methods

### 2.1. Property Distribution Representation

For this discussion, we focus primarily on the 1D finite element situation with linear elements and linear basis functions; however, the points made here are fully generalizable to the 2D and 3D cases where the elements could include various shapes including triangles and tetrahedrons and for linear and more complicated basis functions. We also focus on the electromagnetic field calculation application. For finite element modeling, there are two primary ways to define the property distribution prior to running the actual simulation. The first is by assigning the properties as a function of each linear element. The attraction for this approach is that you can accurately represent the property distribution in a piecewise manner depending on your discretization requirements. The properties are constant over an entire element and jump at the boundaries (i.e., the nodes) depending on the properties of the adjacent element. This has appeal for cases where the domain can be divided into discrete regions with different properties—the human body is a good example where the different internal organs generally have different physical properties. This is how the data is organized in the University of Wisconsin MR breast image repository [20] except that it uses a finite difference time domain grid instead of a finite element mesh.

However, this configuration is not always practical for different modeling situations. For instance, in the image reconstruction configuration used by our group, we utilize the nodal adjoint method for computing the Jacobian matrix during the imaging process. The adjoint method has shown to be a significant improvement over alternate ways of computing the Jacobian, and recent innovations have further improved the computation time to the point that that it is no longer a time limiting computation factor [30]. A key feature in this scenario is that the properties need to be represented at the nodes and not the elements. For this scenario the properties vary linearly across the element with different values defined at each of the nodes. With respect to modeling purposes, the associated stiffness matrices are constructed as an integration process over all the elements. The contributions from each node are computed as part of weak form integrations as a preliminary step towards computing the associated field distributions—which are calculated at each node. The challenge in this situation is that the properties are no longer stepwise constant across the elements. A case can be made that this configuration approximates a stepwise change from one node to the next at their midpoint. It approaches a realistic interpretation as the length of element β shrinks (Figure 1).

Where this issue imparts particular relevance is when modeling the skin layer in the body for applications such as heating via microwave hyperthermia and microwave imaging. Typical thicknesses of the skin are generally of the order of 1.5 mm. When representing an object such as an entire female breast, utilizing a uniform grid with such a small grid element size results in a very large forward computational problem. Even with non-uniform element discretization, the 1.5 mm skin thickness implies that the skin would be represented as essentially a single element layer between the subcutaneous fat layer and the outside coupling medium. Figure 1 shows a 1D representation of the property distribution transecting through the subcutaneous fat layer, through the skin and into the bath. In this case, the property values of the nodes on both sides of element β are set to published values for skin [2], those for the nodes to the left of i to the values for fat and those to the right of j to the values we’ve used for associated imaging coupling baths [31]. For economy purposes, only the distributions of the relative permittivity are shown. From an intuitive perspective, assigning skin properties to nodes i and j, i.e., the points bracketing element β, is reasonable. However, as can be seen from Figure 1, it is clear that the influence of setting the property values at these nodes to those of the actual skin values extends beyond just that of element β. The properties also defined at node i influence the values within element α and those at node j influence the properties within element γ. In the situation where elements α and γ are roughly the same length as β, a case can be made that the effective influence of the skin is actually closer to twice the desired skin thickness. Clearly this problem does not arise when the properties are assigned in an element-based manner.

One strategy is to shrink the skin layer thickness by a factor of two [32]. This is most effective when the modeled layer of elements on either side of element β are roughly the same size as β. While it is possible to substantially increase the size of the elements on both sides of β, in practice, this is generally not done—especially for the more common 2D and 3D simulation problems. For the quality of the simulation, it is important that the elements be relatively normally shaped, i.e., no vertices with excessively sharp angles. This ensures that the element sizes do not change overly abruptly as one progresses away from the skin layer. Figure 2 shows the original distribution from Figure 1 along with the new distribution. Alternatives to this strategy exist, including assignment of skin properties only to nodes on the tissue surface as in these simulations. The important point is to quantify the simulated influence of the skin in comparison to the actual skin thickness.

### 2.2. Finite Difference Time Domain Forward Solution

While the representation of the skin described above is for a finite element-based configuration, the actual forward solutions are computed using finite difference time domain (FDTD). This poses two challenges which need to be balanced against one another. First, the finite element discretization is ideal for non-uniform shapes such as human tissue while FDTD utilizes a uniform grid which can only produce stepwise conformation to the actual shapes. The smaller the FDTD discretization, the more closely it resembles the actual distribution. However, the overall problem size presents a practical limit on how small a grid spacing can be achieved. For this situation, we were able to reduce the grid spacing to 0.83 mm which was nominally 70 nodes per wavelength—well above our usual criterion of 10 nodes per wavelength. More importantly for this situation, we explore results for cases where the skin layer is on the order of 1.5 mm thick. The overall size of the FDTD grid is 18.93 cm × 18.93 cm × 8.51 cm for a rough total of 5.5 million grid points (230 × 230 × 104).

While our preference for the actual implementation of the finite element mesh would be to configure multiple node layers at the skin, it is impractical to construct a mesh with such fine resolution given the current state-of-the-art in computing power. Instead, we have opted to assign the properties of just the nodes on the surface of the breast mesh to published values for skin. The finite element mesh represents just the breast with no nodes or elements outside of that geometry. Our custom mesh generator is unable to provide direct control of the element sizes for those covering the surface. To estimate the outer layer thickness of the mesh, we have taken an average of the heights of all the surface tetrahedra that have a full triangle on the breast surface. That triangle becomes the effective base of the tetrahedron, and the height is the perpendicular distance to the fourth node. The custom mesh generation software package was designed to provide tetrahedral elements that are as close to equilateral tetrahedra as possible. Because of this, our height measurement technique provides a reasonable estimate of the distance from the outer surface to the next layer within the mesh. It should be noted that multiple breast meshes were generated until we found one with suitable skin thickness characteristics.

For the case considered in this analysis, the average tetrahedron height is on the order of 2.81 mm. This is useful in that the 0.83 mm sampling from the overlying FDTD grid provides roughly three samples from the breast surface to the first interior layer of finite element nodes. As an example of this, Figure 3a shows a transect along an FDTD grid line (after mapping from the finite element mesh to the FDTD grid—discussed below) extending from the coupling bath, through the breast and back to the coupling bath. Figure 3b shows a close up of the plotted property features at the breast surface. The peak occurs where the grid sampling occurred closest to the mesh surface. The values taper nearly linearly towards the internal portion of the breast with the last section before reaching the internal breast properties slightly shallower than the other sections. For this particular tetrahedral element through which the associated grid line transects, we can estimate its height as that from the peak and interpolating where the larger sloped length would intersect the breast properties level. In this case, the height is 2.59 mm. This is consistent with our overall average estimate of the outer layer height of 2.81 mm. For this portion of the skin, the effective thickness of the skin is roughly 1.295 mm after taking into account for the factor of 2 required because the associated tetrahedron skin basis functions tapers from 1 at the surface to zero along that length.

With respect to mapping the properties from the FE to FDTD domains, we operate on each FDTD grid point individually. We first determined the FE element in which the grid point resides and then evaluate the four (i.e., for a tetrahedron) linear basis functions at that point. These are then used to compute the properties at the point by weighting the values at the tetrahedron vertices by the associated basis functions and summing their products. This property value distribution is then used to run the FDTD forward solutions. Equation (1) describes the calculation of the properties at an arbitrary point within a tetrahedral element.
(1)$$P\left(x,y,z\right)=\overset{4}{\underset{j=1}{\sum }}P_{j}\phi_{j}\left(x,y,z\right)$$
where *x*, *y*, and *z* are the coordinates of the arbitrary point within the element, *ϕ**_j_* are the element basis functions evaluated at the point, and *P_j_* are the property values at the four element vertices. Each basis function varies linearly from unity at their respective vertex to zero along the opposing surface. This process is repeated for all FDTD grid points and can be readily and efficiently implemented as a matrix operation.

It should be noted that the manner in which the mapping of the property distribution from the finite element mesh to the FDTD grid does contribute an additional non-zero length to the overall skin thickness from outside the breast. By virtue of how the mesh is sampled by the FDTD grid points, the peak property point essentially designates the breast boundary. However, the value for next sample point extending away from the breast mesh is that of the bath properties. Similarly to how the basis functions operate in finite elements, the FDTD-based property distribution decreases linearly from the peak down to the bath properties over one grid spacing. This essentially means that the effective skin thickness increases by one half of an FDTD grid sampling spacing. Because of this, we define the skin thickness as one half of the sum of the average surface tetrahedra heights plus the FDTD grid spacing. For this particular configuration, that becomes 1.71 mm. While this is slightly larger than our stated goal of 1.5 mm, it is sufficiently close to provide a representative assessment of the effect of including the skin in our forward model.

## 3. Results

A 1300 MHz continuous wave (CW) signal was used in all cases. Our imaging system typically acquires data from 700 MHz to 1700 MHz in 200 MHz increments using a low (10 Hz) IF bandwidth. To reduce acquisition time, data are often measured at a few frequencies—six in the examples reported here. We do not reconstruct images at lower frequencies (they are low in resolution), but the data are necessary for phase unwrapping (see [4]). Higher frequency images are generated, but they become susceptible to noise-related artifacts as signals approach the dynamic range limits of the system. We have found images recovered at 1300 MHz represent a good balance between image resolution (higher frequency) and measurement reliability (lower frequency).

### 3.1. Field Distributions

While FDTD-based property maps exist for actual patient breasts in the Wisconsin data base [20], we utilize a breast MRI of a woman who had a MRI performed while her breast was suspended in a glycerin:water coupling medium instead of air because the breast is buoyant in liquid which tends to compress it towards the chestwall relative to when pendant in air.

The breast MRI was segmented manually and transformed into a 3D finite element mesh (Figure 4). Because the skin layer is so thin and not well represented in the MRI, segmentation focused only on adipose and fibroglandular tissues. Using our custom mesh generation software, we controlled the number of finite element nodes and their density [33]. We added a skin layer after segmentation by applying the skin properties to nodes on the boundary of the breast. In this way, the influence of the skin properties (and subsequently its effective thickness) was a function of the average spacing between nodes on the boundary and those one layer internal from the boundary. The overall mesh was comprised of 10,502 nodes and 50,949 elements which corresponded to an average nodal spacing of 2.6 mm. The finite element-based property distribution was mapped to an FDTD distribution for calculating the field solutions. Figure 5a,b show plots of the permittivity distribution for planes through the breast in the FDTD model with and without skin, respectively.

Figure 6a–c show 1300 MHz horizontal field magnitude and phase distributions (anatomically coronal slice through a single plane of the breast) with and without skin, and for the homogeneous bath, respectively. Electrical properties of the 80% glycerin bath were ε_r_ = 25.4, and σ = 1.44 S/m, respectively. The source location and perturbations near the breast periphery are evident. However, differences between the microwave field distributions with and without skin are not perceptible.

Figure 7 shows normalized (a) magnitude and (b) phase values with and without skin at the associated receive antenna locations in the same plane as the transmitter (top—plane 1, bottom—plane 2). Data in Figure 7 are calibrated through a process in which measured magnitudes and phases are subtracted from the same data acquired in a homogeneous bath (with no breast or phantom). Accordingly, 0 dB means no magnitude difference exists in data acquired when the object is and is not present. Similarly, 0 degrees represents no phase difference in the subtracted data. For the magnitude graphs, differences between the skin and non-skin cases are small, and mostly visible for the receive antennas physically furthest from the transmitter (relative receive antennas 1 and 15 are adjacent to the transmitter and the remaining ones are numbered circumferentially such that #8 is furthest from the transmitter). However, even these differences are only 0.61 and 0.82 dB in the worst cases, respectively, with average differences (over all receive antennas) between skin and non-skin cases of 0.20 and 0.40 dB, respectively. The situation is similar for phase graphs where the largest differences occur near antenna 8, and have maximums of 14.1 and 21.5 degrees and averages of 6.87 and 12.2 degrees, respectively.

### 3.2. Reconstructed Images

Figure 8a,b show the 1300 MHz 2D reconstructed permittivity and conductivity images for the simulated phantom at two horizontal planes for both the skin and non-skin cases. Here, we utilized our 2D Gauss-Newton iterative algorithm with a logarithmic transformation for minimizing the variance during convergence [34]. The algorithm is finite element based and utilizes a fine mesh for the forward solution computed at each iteration and a superimposed parameter mesh for reconstructing the image (generally more coarse than the forward solution mesh). We extracted the e_z_ component of the simulated measurement data for the 2D imaging problems. Noise with an amplitude of −100 dBm was added (the transmit power level is 0 dBm) [4]. Synthetic data were generated from 16 antennas, each operating as the transmitter with the complement of 15 antennas acting as receivers (total of 240 measurements). Antennas were positioned uniformly on a 15.2 cm diameter circle with a concentric imaging zone of 14.5 cm diameter. The coupling bath was an 80:20 mixture of glycerin and water with permittivity and conductivity of 25.4 and 1.44 S/m, respectively. The starting property image estimate were values of the coupling bath. The algorithm utilized a standard Levenberg–Marquardt regularization scheme to produce a smoothed image as a starting estimate for a second reconstruction stage which applied a Tikhonov regularization with a Euclidean distance penalty term to produce a more refined result. The algorithm converged in fewer than 30 iterations—typically in less than 3 min. The perimeter of the breast is evident in the reconstructions, especially in the permittivity maps, and properties in the background are uniformly greater than the breast. Some artifacts appear—more prominently in the conductivity images—which is typical of our imaging algorithm and is associated in part with the fact that the forward solutions are calculated in 3D while the reconstructions are performed in 2D. Because the breast was compressed due to buoyancy in glycerin, it appears somewhat flat near the nipple. Correspondingly, permittivity images indicate the breast is disappearing towards planes 6 and 7. Elevated property zones correspond to the fibroglandular tissue in planes 1 and 2. Overall minimal differences occur between the two image pairs as expected since the input measurement data were so similar.

Figure 9 shows difference images between reconstructions with and without the skin layer (Figure 8a,b). A ring of elevated permittivity appears where the skin layer is located (the color scale has been compressed to accentuate the small increase). The corresponding conductivity difference images exhibit similar elevated property rings for some planes but with artifacts in the center of the field of view. Overall, differences are small, but the algorithm is sufficiently sensitive to recover these features even when the associated scattered field input measurements demonstrate very little change.

## 4. Discussion and Conclusions

Differences are small for both forward solutions and reconstructed images with and without skin relative to radar approaches based on reflection techniques. Reflection-based systems operate with a nearly lossless coupling medium to ensure maximum signal penetration into the deeper portions of tissue, and as a result, reflected signals from property mismatch between the medium and skin remain high.

In contrast, our imaging system utilizes a lossy coupling medium which suppresses signal perturbations at discontinuities, and its transmission measurements are less prone to large fluctuations during propagation through the medium and tissue. The test cases developed here reasonably reflect conditions encountered in clinical breast imaging cases. Perturbations in transmission signals do occur because of the presence of skin; however, they are minor and cause minimal changes in recovered images.

When examining difference images for cases with and without skin, the layer is visible, suggesting that recovery of small and subtle features is possible. However, we have not observed the skin as part of clinical imaging exams except in cases where its thickness is pronounced. For example, women with large tumors that have involvement near the breast surface often present with thickening of the skin. In these instances, we would expect large dielectric property increases since the thickening is primarily due to increased edema—i.e., a large influx of saline. Increased permittivity and conductivity arcs around the breast have been visible in these situations [12]. The thickened skin diminished as treatment progressed, and so did the permittivity and conductivity rings in the images.

Most clinical microwave images of the breast and other anatomical sites, for instance the brain, have been acquired with coupling liquids or gels to suppress unwanted multi-path signals [4], Micrima [14], Medfield Diagnostics [35], and EMTensor [36]. Under these conditions, results presented here demonstrate that the skin has minor impact on reconstructed images. While incorporating the skin in field calculations for simulating the image reconstruction process is certainly useful, omitting causes relatively minor differences on internal features, which continue to be recovered with good fidelity. Further, including the skin adds to the overall computation time. Thus, for transmission-based tomographic imaging approaches, reconstruction results are reasonably well served without including skin in field solutions.

## Figures and Tables

**Figure 1 sensors-22-07353-f001:**
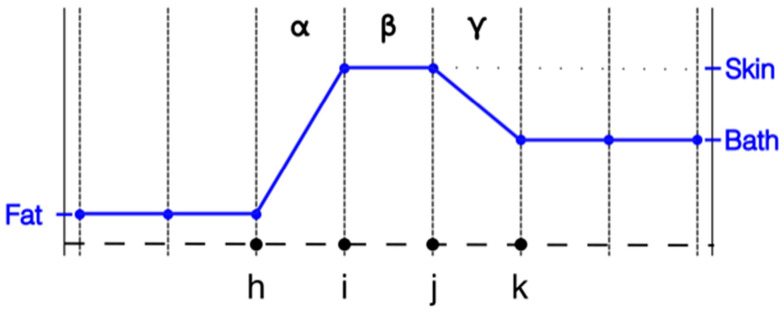
1D property distribution transecting from fat to skin to coupling bath, respectively.

**Figure 2 sensors-22-07353-f002:**
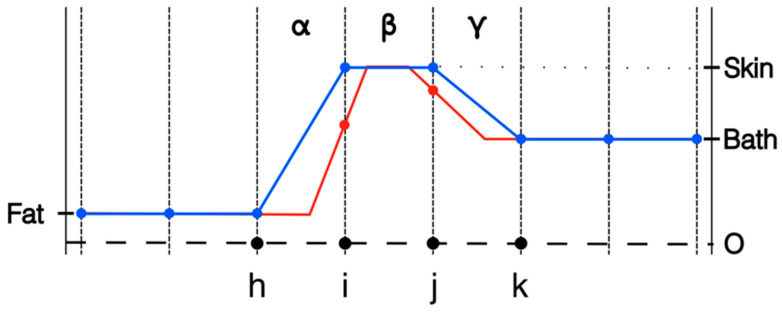
1D representations of the transecting property distributions from fat to skin to coupling bath using nodal representation: full element width, and half element width, respectively.

**Figure 3 sensors-22-07353-f003:**
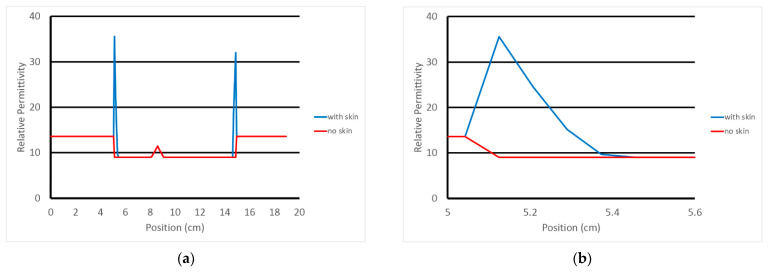
Graph of the relative permittivity along a single FDTD grid line through the breast: (**a**) transect across full domain, and (**b**) close-up in the vicinity of the left hand skin intersection, respectively.

**Figure 4 sensors-22-07353-f004:**
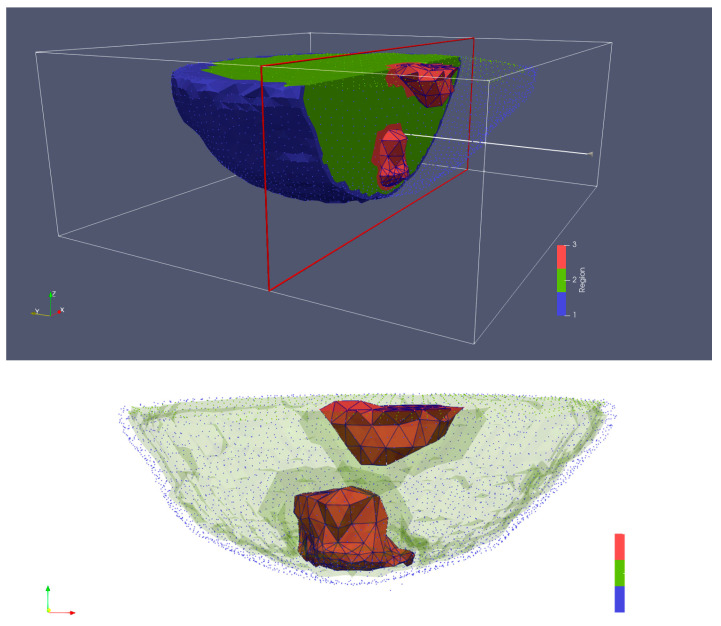
3D finite element mesh generated from an MRI of an actual human breast suspended in a glycerin:water bath—blue for skin, green for adipose tissue, and red for fibroglandular tissue.

**Figure 5 sensors-22-07353-f005:**
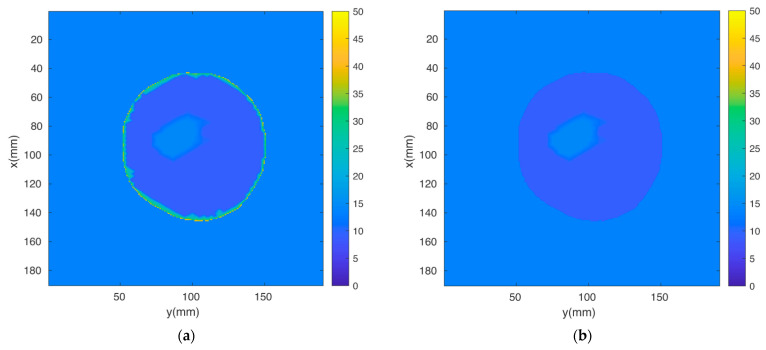
Two-dimensional FDTD permittivity distributions through the breast for cases (**a**) with and (**b**) without skin.

**Figure 6 sensors-22-07353-f006:**
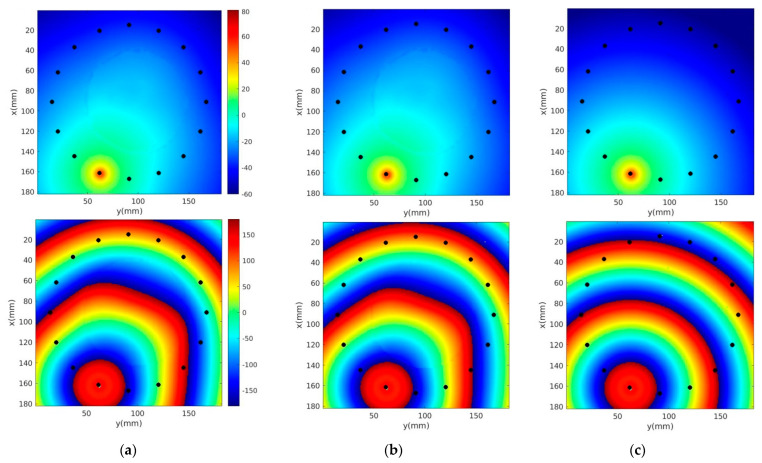
1300 MHz E_z_ magnitude (**top**) and phase (**bottom**) distributions of a single horizontal plane for the cases (**a**) with and (**b**) without skin, and (**c**) the homogeneous bath. Black dots correspond to antenna locations.

**Figure 7 sensors-22-07353-f007:**
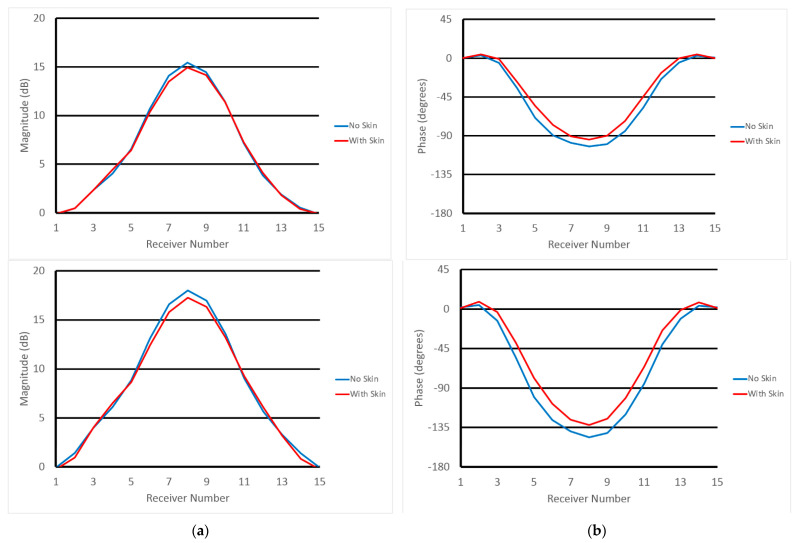
1300 MHz calibrated (**a**) magnitude and (**b**) phase values at relative receive antenna locations for skin and non-skin cases—for the two planes closest to the chestwall (**top** row is closest to the chestwall and **bottom** row is 1 cm below).

**Figure 8 sensors-22-07353-f008:**
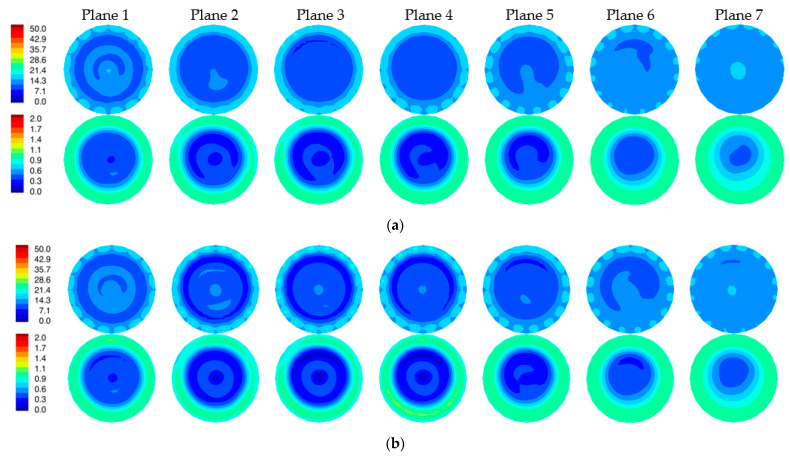
Reconstructed 2D 1300 MHz permittivity (**top**) and conductivity (**bottom**) images for all seven planes of the simulated breast phantom for the cases with (**a**) and without (**b**) the skin layer, respectively.

**Figure 9 sensors-22-07353-f009:**
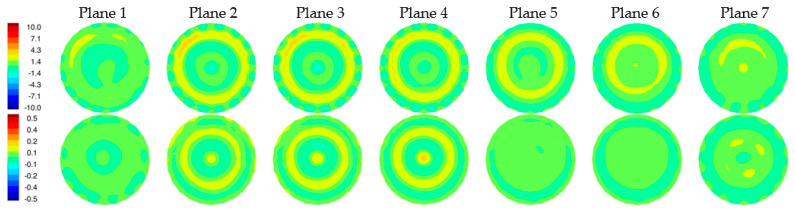
Difference images of reconstructions with and without skin shown in Figure 8 (permittivity images on top row and conductivity images on the bottom row).

## Data Availability

Data is available upon request.

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
