# Peer review of "Impact of Skin on Microwave Tomography in the Lossy Coupling Medium"

_sensors, 2022, doi:10.3390/s22197353_

Round 1

Reviewer 1 Report

It is an interesting work on the skin effect based on transmission-based technology. This technology could be helpful for breast cancer research.

I have two questions here:

1. There are two curves on Figure 2. Is red curve the one used for simulation and blue curve the real property distribution? If yes, how do you validate the impact on the modeling using a different curve quantatively?  

2. On the Discussion and Conclusion Section, you mentioned that permittivity and conductivity ring will thin down as the thicken skin diminished. And you also mentioned that skin is not part of clinical exam if thickness is not pronounced. Is that possible to use your modeling to see how the permittivity/conductivity influenced by skin thickness? 

Author Response

Dear Reviewers,

Thank you for the useful comments and questions.  We have addressed them below and think this has improved the manuscript.

It is an interesting work on the skin effect based on transmission-based technology. This technology could be helpful for breast cancer research.

Thank you.

  1. There are two curves on Figure 2. Is red curve the one used for simulation and blue curve the real property distribution? If yes, how do you validate the impact on the modeling using a different curve quantatively?  

The intent of Figure 2 is to show that neither distribution is exact given constraints of the approach.  As noted in the text, if distance from node i to node j is close to the thickness of the skin, the blue curve overestimates its influence.  Given the added influence that comes from the skin contributions in neighboring elements a and g, shortening the length of element b (red curve) is one way to compensate for the effect whereas the intuitive approach of setting the properties of the two nodes (i and j) to skin values exaggerates the effects of the thickness of the skin.  Applying properties at nodes is viable and is used in situations where property distributions vary continuously over a domain [Huebner et al 2001]. The graphs in Figures 3-5 illustrate the situation for this particular skin implementation.  We have re-written and added text to illustrate the two approaches and emphasize that the latter is valid and in line with how we have treated the skin in our results.

Huebner KH, Dewhirst DL, Smith DE, Byrom TG, The finite element method for engineers, Fourth Edition, pp. 85-108, John Wiley and Sons, New York, 2001.

  1. On the Discussion and Conclusion Section, you mentioned that permittivity and conductivity ring will thin down as the thicken skin diminished. And you also mentioned that skin is not part of clinical exam if thickness is not pronounced. Is that possible to use your modeling to see how the permittivity/conductivity influenced by skin thickness? 

The skin thickening example we refer to in the Discussion and Conclusion section occurred because of tumor progression during neoadjuvant chemotherapy, and has not been observed in our experience during diagnostic breast imaging.  The purpose of the paper is exactly as the reviewer requests – “…to see how the permittivity/conductivity influenced by skin thickness.”. Investigating the effects of exaggerated skin thickening that can occur during neoadjuvant chemotherapy monitoring is a worthy study, but one better suited to a paper focused on therapy monitoring.

Reviewer 2 Report

Dear Respected Authors,

I have found your paper very interesting.

For purposes of reviewing process I ask you to comment and modify the paper according to the following issues:

1. References: I think that there is too much self-citations of Mr. Meaney.

2. Please explain for the readers why do you use frequency of 1300MHz. What is the type of sounding signal: CW or a pulse ?

3. What about the use of wideband signals, for example when the bandwidth is enough large to obtain range resolution equal to the skin thickness ?

4. Fig. 7 - what is the reference for results in dB ?

Best regards

Author Response

Dear Reviewers,

Thank you for the useful comments and questions.  We have addressed them below and think this has improved the manuscript.

I have found your paper very interesting.

Thank you.

For purposes of reviewing process I ask you to comment and modify the paper according to the following issues:

  1. References: I think that there is too much self-citations of Mr. Meaney.

We have removed two of the references to our own work.  Further deletions would compromise key points.

  1. Please explain for the readers why do you use frequency of 1300MHz. What is the type of sounding signal: CW or a pulse ?

Our system typically acquires data from 700 MHz to 1700 MHz in 200 MHz increments using a low IF bandwidth. To reduce acquisition time, data are often measured  at a few frequencies – six in this case.  We do not reconstruct images at lower frequencies (they are low in resolution), but the data are necessary for phase unwrapping (see [4]).  Higher frequency images are generated, but they become susceptible to noise-related artifacts as signals approach the dynamic range limits of the system.  We have found images recovered at 1300 MHz represent a good balance between image resolution (higher frequency) and measurement reliability (lower frequency).

The data we use result from CW signals sampled over a designated time interval. 

  1. What about the use of wideband signals, for example when the bandwidth is enough large to obtain range resolution equal to the skin thickness ?

The idea is interesting, and in fact, the Micrima and Medfield Diagnostics systems acquire data over a wide range.  However, they utilize a much lower dynamic range.  In our approach, a lossy coupling bath suppresses multi-path signals but requires measurements over a very large dynamic range which limits the accessible imaging data bandwidth to  about 900 – 1500 MHz –too narrow to measure a wide-band response.   

We have reconstructed multi-frequency images which are analogous to wideband reconstructions, but only for a few frequencies at a time, and our system and algorithms are not equipped to acquire/process more refined, broadband sampling.

Notably, the Micrima system collects data over the 3 – 10 GHz frequency range, and its combination of a higher and broader frequency range may achieve better skin recovery, although we are not aware of any published data illustrating this feature.

  1. Fig. 7 - what is the reference for results in dB ?

Data in Figure 7 are calibrated through a process in which measured magnitudes and phases are subtracted from the same data acquired in a homogeneous bath (with no breast or phantom). Accordingly, 0 dB means no magnitude difference exists in data acquired when the object is and is not present.  Similarly, 0 degrees represents no phase difference in the subtracted data.   We added a clarifying statement to the paper.